# MERGING MODELS PRE-TRAINED ON DIFFERENT FEATURES WITH CONSENSUS GRAPH

## ABSTRACT

Learning an effective global model on private and decentralized datasets has become an increasingly important challenge of machine learning when applied in practice. Federated Learning (FL) has recently emerged as a solution to address this challenge. In particular, the FL clients agree to a common model parameterization in advance, which can then be trained collaboratively via synchronous aggregation of local model updates. However, such a strong requirement of modeling homogeneity and synchronicity across clients makes FL inapplicable to many practical scenarios. For example, in distributed sensing, a network of heterogeneous sensors sample from different data modalities of the same phenomenon. Each sensor thus requires its own specialized model. Local learning therefore happens in isolation but inference still requires merging the local models to achieve consensus.

To enable isolated local learning and consensus inference, we investigate a feature fusion approach that extracts local feature representations from local models and incorporates them into a global representation for holistic prediction. We study two key aspects of this feature fusion. First, we use alignment to correspond feature components which are arbitrarily arranged across clients. Next, we learn a consensus graph that captures the high-order interactions among data sources or modalities, which reveals how data with heterogeneous features can be stitched together coherently to achieve a better prediction. The proposed framework is demonstrated on four real-life data sets including power grids and traffic networks.

## 1 INTRODUCTION

To improve the scalability and practicality of machine learning applications in situations where training data are becoming increasingly decentralized and proprietary, Federated Learning (FL) (McMahan et al., 2017; Yang et al., 2019a; Li et al., 2019; Kairouz et al., 2019) has been proposed as a new model training paradigm that allows data owners to collaboratively train a common model without having to share their private data with others. The FL formalism is therefore poised to resolve the computation bottleneck of model training on a single machine and the risk of privacy violation, in light of recent policies such as the General Data Protection Regulation (Albrecht, 2016).

However, FL requires a strong form of homogeneity and synchronicity among the data owners (clients) that might not be ideal in practice. First, it requires all clients to agree in advance to a common model architecture and parameterization. Second, it requires clients to synchronously communicate their model updates to a common server, which assembles the local updates into a global learning feedback. This is rather restrictive in cases where different clients draw observations from a different data modality of the phenomenon being modeled. It leads to heterogeneous data complexities across clients, which in turn requires customized forms of modeling. Otherwise, enforcing a common model with high complexity might not be affordable to clients with low compute capacity; and vice versa, switching to a model with low complexity might result in the failure to unlock important inferential insights from data modalities.

A variant of FL (Hardy et al., 2017; Hu et al., 2019; Chen et al., 2020), named vertical FL, has been proposed to address the first challenge, which embraces the concept of vertically partitioned data. This concept is figuratively named through cutting the data matrix vertically along the feature axis, rather than the data axis. Existing approaches generally maintain separate local model parameters

distributed across clients and global parameters on a central server. All parameters are then learned jointly, causing however a practical drawback:

**Coordination overhead among clients and the central server, such as engineering protocols that enable multiple rounds of communication (i.e., synchronicity) and coordination effort (i.e., homogeneity) to converge on universal choices of models and training algorithms, would be required, which can be practically expensive depending on the scale of the application.**

To mitigate both constraints on homogeneity and synchronicity[1] satisfactorily, we ask the following question and subsequently develop an answer to it:

**Can we separate global consensus prediction from local model training?**

As shown later in our experiments, we will address this question in a real-world context of the national electricity grid, over which thousands of phasor measurement units (PMUs) were deployed to monitor the grid condition and data were recorded in real-time by each PMU (Smartgrid.gov). PMU measurements, as time series data, are owned by several parties, each of which may employ different technologies leading to heterogeneous recordings under varying sampling frequencies and measured attributes. These data may be used to train machine learning models that identify grid events (e.g., fault, oscillation, and generator trip). Such an event detection system relies on collective series measurements at the same time window but distributed across owners. Using VFL to build a common model on such decentralized and heterogeneous data is plausible but not practical, because of a lack of autonomy that facilitates coordination among the owners.

To resolve the challenge, we instead introduce a feature fusion perspective to this setting, which aims to minimize coordination among clients and maximize their autonomy via a local–global model framework. Therein, each client trains a customized local model with its data modalities. The training is independent and incurs no coordination. Once trained, local feature representations of each client can then be extracted from the penultimate layer of the corresponding local models. Then, a central server collects and aggregates these representations into a more holistic global representation, used to train a model for global inference. There are two technical challenges that need to be addressed to substantiate the envisioned framework.

**C1. There is an ambiguity regarding the correspondence between components of local feature representations across different clients.** This ambiguity arises because local models were trained separately in isolation and there is no mechanism to enforce that their induced feature dimensions would be aligned. As a matter of fact, it is possible to permute the induced feature dimensions without changing the prediction outcome. Thus, if two models are trained separately, they might end up looking at the same feature space but with permuted dimensions.

**C2. There are innate local interactions among subsets of clients that need to be accounted for.** Naively concatenating or averaging the local feature representations accounts for the global interaction but ignores such local interactions, which are important to boost the accuracy of global prediction as shown later in our experiments.

To address **C1**, note that the feature dimension alignment problem is discrete in nature; furthermore, there is no direct feedback to optimize for such alignment. To sidestep this challenge, we develop a neuralized alignment layer whose parameters are differentiable and can therefore be part of a larger network, including the feature aggregation and prediction layers, which can be trained end-to-end via gradient back-propagation (Section 4). To address **C2**, we employ graph neural networks as the global inference model, where the graph corresponds to the explicit or implicit relational structure of the data owners. As such a graph might not be given in advance, we treat the combinatorial graph structure as a random variable of a product of Bernoulli distributions whose (differentiable) parameters can also be optimized via gradient-based approach (Section 5). The technical contributions of this work are summarized below.

**1.** We formalize a feature fusion perspective for distributed learning, in settings where data is vertically partitioned. This is an alternative view to VFL but as elaborated above, is more applicable when iterative training synchronicity is not possible among clients (Section 2).

---

[1]Note that in our case, synchronicity requires co-training among clients which is a weaker constraint than its usual meaning of further requiring clients to synchronize their updates per iteration.

**2.** We formulate a federated feature fusion (F$^3$) framework that consists of a network of pre-trained local models and a central model that collects and fuses the local feature representations (induced from these pre-trained models) to generate a global model with better predictive performance (Section 3). This is achieved via addressing **C1** (Section 4) and **C2** (Section 5) above.

**3.** We demonstrate experiments with four real-life data sets, including power grids and traffic networks, and show the effectiveness of the proposed framework (Section 6).

## 2 PROBLEM SETTING AND RELATED WORK

Federated Feature Fusion (F$^3$) is a new but more practical setup for VFL (Hu et al., 2019; Chen et al., 2020); it aims to enable collaboration between data owners that possess private access to different sets of features describing the same set of training data points. However, unlike VFL which require clients to synchronize their training processes (Yang et al., 2019b; Li et al., 2021; Fu et al., 2021; Cheng et al., 2021; Hu et al., 2019; Diao et al., 2021) in multiple iterations of communication, F$^3$ allows data owners to train their own local models in isolation and only requires one round of communication in which local feature representations induced from the heterogeneously pre-trained local models are shared with a trusted server for feature fusion.

**Remark.** Previously, similar ideas on extending federated learning to accommodate clients with heterogeneous models (Tan et al., 2022b;a; Lin et al., 2020; Chen et al., 2022) has been proposed but are still restricted to horizontal settings: Local models still need to operate on the same feature space and cannot be trained in isolation which consequently require multiple rounds of communication and potentially incur extra coordination overhead.

Thus, to emphasize on the novelty of our setting and solution significance, we further review and discuss the formulation of VFL and F$^3$ below, which argues with concrete, real-life examples why the F$^3$ setting is more practical and how this practicality would entail significant technical challenges that necessitate new solutions in Sections 4 and 5.

**Federated Learning with Vertically Partitioned Data.** From a data perspective, the decentralized nature of data in VFL is a transposition to that of the traditional horizontal federated learning (HFL) (McMahan et al., 2017). Instead of owning the same set of features for different sets of data points as in HFL, the data owners in VFL now own different sets of features for the same set of data points; and they share a common label set of these data points.

From the existing literature, two lines of work are noted. One takes the data matrix literally – by assuming tabular data and studying linear models – where model parameters have natural correspondence to the data parts (Hardy et al., 2017; Nock et al., 2018; Heinze et al., 2014; 2016). Often, these approaches are hard to generalize to complex data with many owners. Another line of work advocates the use of models with modular structure in which separate parts of the model are responsible to locally aggregate different sets of local features owned by different owners; and a global parameterization is used to combine these local features. This is similar in spirit to F$^3$ but require clients to synchronize the training processes of their assigned model parts, which incurs expensive communication and creates dependence among the clients (Hu et al., 2019; Chen et al., 2020).[2]

Mathematically, for each datum $\mathbf{x}_k$ with label $y_k$, let $\mathbf{x}_k^i$ be the feature set of the datum that the $i$-th owner possesses. That is, $\mathbf{x}_k = (\mathbf{x}_k^1, \mathbf{x}_k^2, \ldots, \mathbf{x}_k^n)$ with $n$ data owners. VFL computes:

$$\underset{\mathbf{w}, \boldsymbol{\theta}}{\text{minimize}} \ \mathbf{L}(\mathbf{w}, \boldsymbol{\theta}) \quad \triangleq \quad \frac{1}{m} \sum_{k=1}^{m} \ell \left[ g\Big( \phi_1\left(\mathbf{x}_k^1; \boldsymbol{\theta}_1\right), \phi_2\left(\mathbf{x}_k^2; \boldsymbol{\theta}_2\right), \ldots, \phi_n\left(\mathbf{x}_k^n; \boldsymbol{\theta}_n\right); \mathbf{w} \Big), y_k \right] \quad (1)$$

where each $\phi_i(\mathbf{x}_k^i; \boldsymbol{\theta}_i)$ is a (learnable) local embedding of $\mathbf{x}_k^i$ parameterized by a separate parameter vector $\boldsymbol{\theta}_i$ owned by the $i$-th owner, $g(\phi_1, \phi_2, \ldots, \phi_n; \mathbf{w})$ is an aggregation function parameterized by $\mathbf{w}$ and $\ell$ is a prediction loss, e.g. the cross-entropy loss for classification or $\ell_2$ loss for regression. The loss in Eq. (1) is averaged over all training data points $\mathbf{x}_1, \mathbf{x}_2, \ldots, \mathbf{x}_m$.

**Federated Feature Fusion.** The setting of F$^3$ is similar to VFL, except that the data owners share neither data nor models with each other to ensure a higher degree of privacy compliance, which is

---

[2]Note that the approach proposed by Hu et al. (2019) assumes no parameters for the global model. Were global parameters present, gradient communication is inevitable.

often the more practical setting in industry – see the example on power grid at the end of this section. For this reason, the VFL minimization task in Eq. (1) above is changed to

$$\underset{\mathbf{w}}{\text{minimize}} \ \mathbf{L}(\mathbf{w}) \ \triangleq \ \frac{1}{m} \sum_{k=1}^{m} \ell \left[ g\left(\mathbf{h}_k^1, \ \mathbf{h}_k^2, \dots, \ \mathbf{h}_k^n; \mathbf{w}\right), y_k \right] \tag{2}$$

where $\mathbf{h}_k^i = \phi_i^*(\mathbf{x}_k^i)$ with $\phi_i^* = \arg\min_{\phi_i} \ell_i\left(\phi_i(\mathbf{x}_k^i), y_k\right)$ which characterizes the locally optimal feature representation obtained in isolation by the $i$-th owner. As such, Eq. (2) only requires one round of communication where $\{\mathbf{h}_k^i\}_{k,i}$ are communicated to a trusted server. Prior to that, each data owner can freely learn their own feature representation model $\phi_i(\mathbf{x}_k^i)$ with different parameterization and architecture, catering towards their own compute capacities and data representation. This avoids forcing the data owners to participate in a joint training scheme which often requires expensive coordination and is not practical. However, in exchange for this practicality, two key challenges arise. First, as local models are separately trained, the correspondence between components of induced feature representations across local models become ambiguous since there is no mechanism to enforce their alignment. Second, for the same reason, there are potential innate local interactions among subsets of clients and a naive concatenation or averaging of their corresponding feature presentations will likely ignore such interactions, resulting in decreasing performance. These correspond to high-level challenge **C1** and **C2** in Section 1 which will be addressed in Sections 4 and 5 as our key technical contributions.

**Data Example.** Let us consider the power grid monitoring task as an example. Figure 1 pictorially illustrates PMU measurements distributed across data owners. A panel of time series corresponds to a specific time window and the series collectively represent one data point, which the event detection system classifies. In this simplified illustration, each data owner possesses one series recorded by one PMU; but in practice they may own different amounts of PMUs (and thus series). Moreover, the series may differ in length because of varying sampling frequencies; and the series are multivariate with possibly different number of variates. All these variations contribute to data heterogeneity, which necessitates the construction of separate local models. Note that if an event does not cascade over the entire grid, some local models may report event whereas others report normal, resulting in conflicting opinions. A consensus global model is responsible to resolve the conflict. Additionally, missing data may occur.

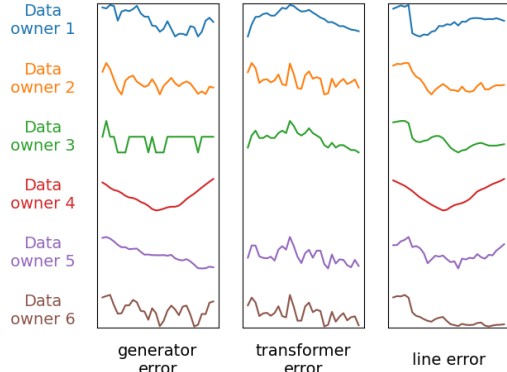

Figure 1: Federated Feature Fusion: A global prediction is produced collectively based on a set of global features which are the result of fusing local feature representations supplied by the data owners. These feature representations are induced from locally trained models on raw local data which might be heterogeneous.

## 3 FEDERATED FEATURE FUSION FRAMEWORK

As detailed above, the proposed framework for Federated Feature Fusion consists of local models $\phi_i$ and a global feature fusion model $g$, such that their composition minimizes the loss in Eq. (2). Each data owner $i$ possesses a local model trained with its data, independently of other owners. This way, no data sharing is invoked and privacy is of minimal concern. However, because the local models lack a global vision and may be conflicting, a central (global) model is key to coordinating the local opinions for final prediction. To maintain autonomy, local models are frozen once pre-trained and will not join the training of the global model. Data owners send local data representations to a centralized server for global model training (and inference). In other words, the global model queries neither the raw data nor the local models from data owners. As long as owners agree to send the less decipherable representations to the central server, global inference can be made.

**Local Models.** We treat a neural network except the final output layer as a feature extractor, which produces the representation $\mathbf{h}_k^i$ of an input fragment $\mathbf{x}_k^i$; and treat for simplicity the output layer as

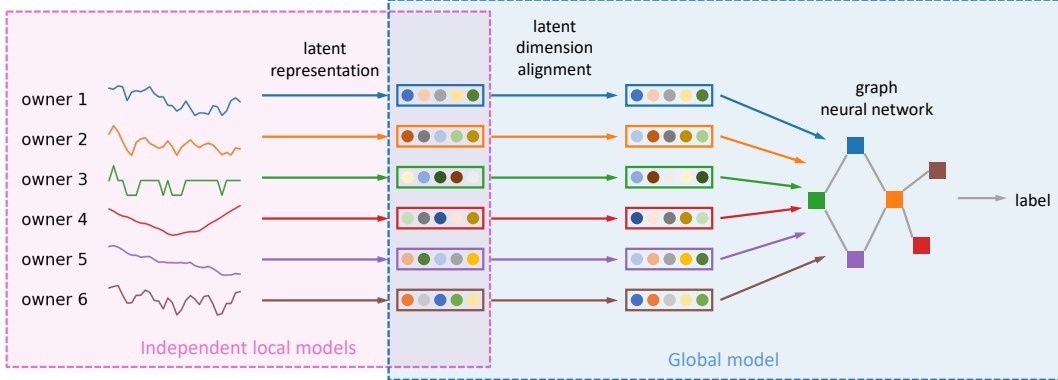

Figure 2: Federated Feature Fusion Framework. Local models are trained independently and separately from the global model. The algorithm is summarized in Algorithm 1.

a logistic regression. That is, a local model $g_i(\mathbf{x}_k^i)$ reads:

$$g_i\left(\mathbf{x}_k^i\right) \triangleq \operatorname{softmax}\left(\mathbf{W}_i \cdot \mathbf{h}_k^i + \mathbf{b}_i\right) \quad \text{where} \quad \mathbf{h}_k^i = \phi_i\left(\mathbf{x}_k^i\right). \tag{3}$$

Hereafter, we will interchangeably use *representation*, *embedding*, and *latent vector* to mean $\mathbf{h}_k^i$. These $\mathbf{h}_k^i$'s are assumed to have the same shape across $i$, although $\mathbf{x}_k^i$ can have different shapes and the embedding function can have different architectures to cope with data heterogeneity. A simple example of the embedding function is a fully connected layer $\mathbf{h}_k^i = \operatorname{ReLU}(\mathbf{U}_i \cdot \mathbf{x}_k^i + \mathbf{c}_i)$; but an arbitrarily complex function is also applicable.

**Global Model.** The global model $g$ melds together all local representations to generate a prediction:

$$y_k \simeq \widehat{y}_k \triangleq g\left(\mathbf{h}_k^1, \mathbf{h}_k^2, \ldots, \mathbf{h}_k^n; \mathbf{w}\right). \tag{4}$$

which is parameterized by $\mathbf{w}$. For example, the parameterization $\mathbf{w} = \{\mathbf{W}_0, \mathbf{W}_1, \mathbf{b}_0, \mathbf{b}_1\}$ can characterize a fully connected layer followed by mean pooling and another fully connected layer:

$$g\left(\mathbf{h}_k^1, \mathbf{h}_k^2, \ldots, \mathbf{h}_k^n; \mathbf{w}\right) = \operatorname{softmax}\left(\mathbf{W}_1 \cdot \frac{1}{n} \sum_{i=1}^n \operatorname{ReLU}\left(\mathbf{W}_0 \cdot \mathbf{h}_k^i + \mathbf{b}_0\right) + \mathbf{b}_1\right). \tag{5}$$

Thus, given a particular parameterization $\mathbf{w}$, we can substantiate Eq. (4) above and plug it into Eq. (2). The optimal value for $\mathbf{w}$ can then be achieved by solving the corresponding minimization task therein. However, designing the form of $\mathbf{w}$ is highly non-trivial and is in fact tied to the previously mentioned challenges **C1** and **C2**, which we further elaborate below.

**Challenges.** Two considerations are pertinent to the design of $\mathbf{w}$. First, when the latent dimensions have semantic meaning – e.g. when the local models are trained to yield disentangled representations (Higgins et al., 2018) – each latent feature of the local representations may not match, because an arbitrary permutation of the latent dimensions does not change a local model. Second, a naive mean pooling as in (5) may miss the interdependencies between local data, leading to a less well performing global model. Such interdependencies naturally occur in the power grid example because of the physics of an electricity network. Hence, in subsequent sections, we use latent alignment to address the first problem and graph neural network to address the second one. Incorporating these two components, we show the full proposed framework in Figure 2 and Algorithm 1. We now discuss the solutions to these challenges in Sections 4 and 5 below.

## 4 Aligning Local Representations

For the global model to be meaningful, the feature dimensions of the local representations $\mathbf{h}_k^i$ should be aligned under the same feature space. For example, in (5), all $\mathbf{h}_k^i$'s multiply the same weight matrix $\mathbf{W}_0$; in other words, each element of $\mathbf{h}_k^i$ corresponds to one input neuron of the initial fully connected layer. Permutations of the elements will destroy the correspondence. That is, even if

---

**Algorithm 1** Federated Feature Fusion ($F^3$)

---

1: **function** TRAINING($\{(\mathbf{x}_k^i, y_k)_{k=1}^m\}_{i=1}^n$)
2:        Each data owner $i$ trains a local model $g^i$ with its local data part $(\mathbf{x}_k^i, y_k)_{k=1}^m$.
3:        Each data owner $i$ sends its local data representations $\{\mathbf{h}_k^i\}_{k=1}^m$ to the central server – Eq. (3)
4:        Central server learns $\widehat{y}_k = g(\mathbf{P}_1\mathbf{h}_k^1, \mathbf{P}^2\mathbf{h}^2, \dots, \mathbf{P}^n\mathbf{h}^n)$ via Eq. (7).
5:        Here, the global model is (8), where the loss is taken over the distribution of $\widehat{A}$.
6:        Entries of $\widehat{A}$ are sampled using (9).
7:        Each alignment matrix $P^i$ is a learnable arbitrary parameter matrix.
8: **end function**

9: **function** INFERENCE($\mathbf{x}_1, \dots, \mathbf{x}_m$ where $\mathbf{x}_k = (\mathbf{x}_k^1, \dots, \mathbf{x}_k^n)$)
10:       Each data owner $i$ evaluates its local model with $\mathbf{x}_k^i$ to obtain $\mathbf{h}_k^i$ and sends to server.
11:       Server evaluates takes $\{\mathbf{h}_k^i\}_{i=1}^n$ as input and produces prediction via Eq. (5).
12: **end function**

---

the local models are fixed, the arbitrary arrangement of the feature dimensions of the latent vectors causes ambiguity of what an optimal global model can be built.

Mathematically, let us use a vector $\mathrm{p}$ to denote the index (column) permutation of a vector (matrix). Then, the $i$th local model (3) can be equivalently written as

$$g_i\left(\mathbf{x}_k^i\right) \quad \triangleq \quad \operatorname{softmax}\left(\mathbf{W}_i\left[:,\mathrm{p}_i\right] \cdot \mathbf{h}_k^i\left[\mathrm{p}_i\right] + \mathbf{b}_i\left[\mathrm{p}_i\right]\right) \quad \text{where} \quad \mathbf{h}_k^i \triangleq \phi_i\left(\mathbf{x}_k^i\right), \quad (6)$$

for any permutation $\mathrm{p}_i$ as long as the embedding function is able to produce a permuted $\mathbf{h}_k^i[\mathrm{p}_i]$ under the same input $\mathbf{x}_k^i$. Such a requirement can be easily satisfied if the embedding function is a fully connected layer such as $\mathbf{h}[\mathrm{p}] = \operatorname{ReLU}(\mathbf{W}[\mathrm{p},:] \cdot \mathbf{x} + \mathbf{b}[\mathrm{p}])$. In fact, it is satisfied by most neural networks as well. In Appendix B, we give another example: the GRU (Cho et al., 2014).

Hence, we propose to align the feature dimensions across all local vectors $\mathbf{h}_k^i$ to disambiguate the ambiguity. This proposal amounts to modifying the global model (4) to the following:

$$y_k \simeq \widehat{y}_k \quad \triangleq \quad g\left(\mathbf{P}_1 \cdot \mathbf{h}_k^1, \mathbf{P}_2 \cdot \mathbf{h}_k^2, \dots, \mathbf{P}_n \cdot \mathbf{h}_k^n\right), \quad (7)$$

where $\mathbf{P}_i$ is an alignment matrix for each data owner $i$, implementing the (manual) index or column permutation above in linear algebra. We can then treat each $\mathbf{P}_i$ as a free parameter matrix to optimize. It may be square or rectangle, the latter case indicating a change of the number of features. We also show an alternative hard alignment by parametrizing $\mathbf{P}_i$ a permutation matrix in Appendix H.

## 5    LEARNING A CONSENSUS GRAPH

The example global model (5) performs a naive averaging for the local representations. Since data owners are often interconnected, a more expressive model exploits their relational interactions to improve inference (Battaglia et al., 2018). To this end, we propose to use a graph neural network (GNN) (Zhang et al., 2020; Wu et al., 2021) to process the latent representations.

**A. Modeling Consensus Graph via GCN with Latent Graph.** Many GNNs are applicable; we focus on GCN (Kipf & Welling, 2017) for its simplicity. Let $\mathbf{A}$ be the graph adjacency matrix and let $\mathbf{H}_k$ be the matrix of aligned local representations:

$$\mathbf{H}_k \quad \triangleq \quad \begin{bmatrix} -(\mathbf{P}_1\mathbf{h}_k^1)^\top - \\ \vdots \\ -(\mathbf{P}_n\mathbf{h}_k^n)^\top - \end{bmatrix}.$$

Traditionally, GCN was designed for node classification so we modify it slightly for our purpose,

$$y_k \simeq \widehat{y}_k \quad \triangleq \quad \operatorname{softmax}\left(\frac{1}{n}\mathbf{1}^\top\widehat{\mathbf{A}} \cdot \operatorname{ReLU}\left(\widehat{\mathbf{A}}\mathbf{H}_k\mathbf{W}_0\right) \cdot \mathbf{W}_1\right), \quad (8)$$

where $\widehat{\mathbf{A}}$ is a normalization of $\mathbf{A}$ – see (Kipf & Welling, 2017) for details – and $\mathbf{W}_0$ and $\mathbf{W}_1$ are weight matrices. The modification is the inclusion of $\frac{1}{n}\mathbf{1}^T$ as pooling before output. Modulo this

modification, the formula (8) is a standard one used in the literature, with the bias terms omitted. It is interesting to note the equivalence of GCN (8) and the graph-agnostic model (5) when $\widehat{\mathbf{A}}$ is replaced by the identity matrix (omitting bias terms).

In GCN, $\mathbf{A}$ corresponds to the consensus graph among local owners as graph nodes. If such a graph is not present, it is possible to learn one such that (8) still outperforms (5). In this case, we treat $\mathbf{A}$ as a random variable of the matrix Bernoulli distribution, where the success probabilities are free parameters to learn. Formally, the elements $\mathbf{A}_{ij}$ are independent and each follows $\mathrm{Ber}(\theta_{ij})$, where $\theta_{ij}$ denotes the corresponding probability (Kipf et al., 2018; Shang et al., 2021). Then, the global model $g$ has $\mathbf{W}_0$, $\mathbf{W}_1$, the $\mathbf{P}_i$'s, as well as $\theta$, as parameters. Following Franceschi et al. (2019); Shang et al. (2021), we formulate the training loss as an expectation over $\mathbf{A}$'s distribution and draws a sample $\mathbf{A}$ to obtain an unbiased estimate of the loss as well as the gradient.

**B. Differentiable Graph Sampling via Re-parameterization.** However, the central challenge of this approach is that the sample $\mathbf{A}_{ij}$ is not differentiable with respect to the corresponding Bernoulli bias $\theta_{ij}$, which in turn makes the training loss non-differentiable with respect to $\theta$. To sidestep this difficulty, we propose the following reparameterization, which presents a learnable (differentiable) transformation of a sample drawn from a continuous distribution to a discrete Bernoulli sample. This transformation is detailed in Definition 1 below, which is followed by Theorem 1 showing the distributional convergence of this transformation to the desired Bernoulli distribution.

**Definition 1.** Let $F$ be the CDF of an arbitrary continuous probability distribution. Sample $s$ from this **reference distribution** and let

$$z \quad \triangleq \quad \mathrm{sigmoid}\left(\frac{1}{\tau}\Big(F^{-1}(\theta) - s\Big)\right), \quad \tau > 0. \tag{9}$$

We call this the **ICDF** re-parameterization which is named after the use of inverse cumulative $F^{-1}$.

**Theorem 1.** For all $\tau > 0$, $\theta \in (0, 1)$ and $t \in [0, 1]$, if the distribution with CDF $F$ is finitely supported on $[a, b]$, then

$$\mathrm{Pr}(z \leq t) \quad = \quad \begin{cases} 0 & \text{if} \quad t < \mathrm{sigmoid}((F^{-1}(\theta) - b)/\tau), \\ 1 & \text{if} \quad t > \mathrm{sigmoid}((F^{-1}(\theta) - a)/\tau), \\ 1 - F(F^{-1}(\theta) + \tau \log(t^{-1} - 1)) & \text{otherwise.} \end{cases} \tag{10}$$

On the other hand, if the distribution is not finitely supported (i.e., $a = -\infty$ and/or $b = +\infty$), Eq. (10) still holds because either (or both) of the first two cases will not be invoked. As a consequence, the distribution of $z$ converges to $\mathrm{Ber}(\theta)$ as $\tau \to 0$.

**Discussion.** We note that an alternative to the above can be achieved via using the Gumbel softmax reparameterization (Jang et al., 2017; Maddison et al., 2017) which also features a differentiable relaxation of the categorical distribution (in this case, the Bernoulli distribution) that approximates it asymptotically. However, in order to obtain one Bernoulli sample, the Gumbel trick requires to sample the Gumbel distribution twice. Instead, our proposed reparameterization only requires sampling from the reference distribution only once. We also show that the ICDF re-parameterization converges as fast as the Gumbel softmax. Both approaches have asymptotic convergence rate on the order of $O(\tau^2)$ as shown in Section D. Empirically, we also show that ICDF induces marginally better performance than Gumbel softmax. This is why we prefer ICDF to Gumbel in our work.

# 6 EXPERIMENTS

In this section, we demonstrate comprehensive experiments to show that federated feature fusion ($\mathrm{F}^3$) can be effectively conducted by using the proposed techniques in Sections 4 and 5.

**Datasets.** We use four real-life, time series datasets. Two are PMU data collected from multiple data owners of the U.S. power grid. For proof of concept, we smooth out heterogeneity and prepare homogeneous data sets. Such a pre-processing is sufficient to test the proposed techniques under minimal impact of the complication by the otherwise diverse local models. Since the PMU data sets are proprietary, we also use two public, traffic data sets (Li et al., 2018) for experimentation. A summary of these data sets is given in Table 1 and the processing details are given in the supplement.

Table 1: Datasets.

|  | METR-LA | PEMS-BAY | PMU-B | PMU-C |
|---|---|---|---|---|
| # Data samples | 2856 | 4343 | 4853 | 1884 |
| # Data owners | 207 | 325 | 43 | 188 |
| Series length | 12 | 12 | 30 | 30 |
| # Features | 1 | 1 | 2 | 2 |
| # Classes | 2 | 2 | 4 | 4 |
| Missing data? | no | no | yes | yes |
| Given graph? | yes | yes | no | no |

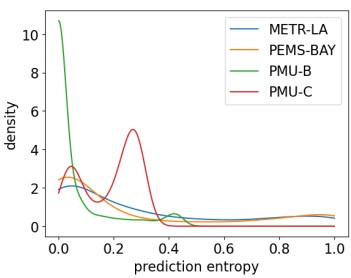

Figure 3: Distributions of prediction entropy across local models.

**Experiment Setting.** All local models are LSTM (Hochreiter & Schmidhuber, 1997) with the same hyperparameters, but pre-trained separately by using local data. The local models are not fine-tuned in the training of the global model. Each dataset is split randomly for training/validation/testing. See the supplement for further details.

**Conflicting Local Predictions.** We first show that local models do not produce consistent predictions, which justifies the effort of training a global model and performing federated inference. For each datum, we compute the entropy of the predicted labels and summarize the entropies for all data into a distribution, plotted in Figure 3. Recall that the lower the entropy, the more consistent the local predictions. The figure, however, shows that a substantial amount of entropies is away from zero, suggesting that local predictions are inconsistent.

**Effectiveness of Federated Feature Fusion.** We make two sets of comprehensive comparisons to evaluate the effectiveness of the proposed framework. The first set, as outlined in Table 2, compares F$^3$ with a number of non-graph baselines (**A–G**) and including (a) a vanilla federated learning baseline (**A**) which requires both model homogeneity and training synchronicity among clients that are not admitted in our setting; (b) a set of standard ensemble strategies (**B–F**) to combine the local models such as voting, binary thresholding, best local model, mean-pooling via Eq. (5), as well as an advanced fusion model via a simplified Set Transformer with 2 layers and 4 heads (Lee et al., 2019) (**F**); and (c) a vertical federated learning baseline (**G**) via feature concatenation. This set also contains several variants of our proposed federated feature fusion model, featuring an ablation study of the effectiveness of our model components: (**H**) F$^3$ without alignment; (**K**) F$^3$ with learnable alignment; and (**J**) F$^3$ with partially tied parameter among local models which require local models to use standard FL to collaboratively learn a common feature aggregation matrix $\forall i \in [n]: \mathbf{W} = \mathbf{W}_i$ – readers can recall the role of $\mathbf{W}_i$ in Eq. (3) – among all local models. Baseline (**K**) is thus an alternative to alignment which comes with the cost of imposing strong homogenization – though not as strong as (**A**) – among local models despite the different nature of their local data. All variants of F$^3$ (**H–K**) use the **ICDF** re-parameterization to learn the graph structure.

From Table 2, we observe that baselines (**A–D**), either lacking necessary localized models or a holistic global model, perform significantly worse than the other baselines (including our F$^3$ variants, the ensemble via mean pooling baseline and vertical federated learning). On the other hand, baselines (**E–H**) perform better than (**A–D**) but lacking a proper alignment of local models or imposing a strong form of homogenization among local models to sidestep alignment, they are expectedly outperformed by baseline (**K**) that performs alignment.

We also compare with two variants of our final model **K** in the vertical federated learning setting. **L** uses the same local and global models as **K** but allows gradients to be sent back to local clients, thus local models can be updated. It achieves similar performance as **K** but leads to much more communication cost with multiple rounds of gradient messages. **M** assumes no local pretrained models and all local and global models are trained jointly from scratch. Its performance is much worse than **K** and **L** – explains the merit of pretrained local models. Another typical VFL baseline (**G**) with pretrained local models and a simple concatenation based global model is also inferior.

**Impact of Learning Graph.** Our next set of experiments, as outlined in Table 3, demonstrate the impact of learning a graph that characterizes the innate local interactions among subsets of clients, following our challenge statement **C2** in the introduction, on both alignment and non-alignment baselines. This provides ablation studies on the isolated impact of having a specific graph learning

Table 2: Effectiveness of latent alignment in a graph-based global model. Superscript numbers are standard deviations. Note that baseline (**A**) and (**H**) are not applicable to the federated feature fusion where (local) model homogenization and training synchronicity are not allowed.

| | METR-LA | | PEMS-BAY | | PMU-B | | PMU-C | |
|---|---|---|---|---|---|---|---|---|
| | F1 | AUC | F1 | AUC | F1 | AUC | F1 | AUC |
| **A**: Federated Learning | $.25^{.000}$ | - | $.33^{.000}$ | - | $.36^{.000}$ | - | $.29^{.000}$ | - |
| **B**: Majority Voting | $.11^{.000}$ | - | $.09^{.000}$ | - | $.29^{.000}$ | - | $.18^{.000}$ | - |
| **C**: Binary Thresholding | $.69^{.000}$ | - | $.64^{.000}$ | - | - | - | - | - |
| **D**: Best Model Selection | $.53^{.000}$ | $.70^{.000}$ | $.55^{.000}$ | $.79^{.000}$ | $.37^{.000}$ | $.69^{.000}$ | $.32^{.000}$ | $.62^{.000}$ |
| **E**: Mean Pooling – Eq. (5) | $.77^{.009}$ | $.96^{.004}$ | $.74^{.012}$ | $.93^{.001}$ | $.38^{.008}$ | $.71^{.006}$ | $.34^{.008}$ | $.64^{.010}$ |
| **F**: Transformer | $.78^{.023}$ | $.94^{.018}$ | $.72^{.045}$ | $.93^{.027}$ | $.39^{.003}$ | $.70^{.009}$ | $.40^{.053}$ | $.67^{.058}$ |
| **G**: Concatenation | $.83^{.008}$ | $.97^{.002}$ | $.80^{.066}$ | $.96^{.028}$ | $.39^{.006}$ | $.71^{.036}$ | $.40^{.025}$ | $.68^{.040}$ |
| **H**: F$^3$ with no alignment | $.80^{.009}$ | $.96^{.004}$ | $.75^{.009}$ | $.94^{.001}$ | $.39^{.003}$ | $.73^{.015}$ | $.40^{.020}$ | $.66^{.018}$ |
| **J**: F$^3$ with parameter tying | $.82^{.009}$ | $.97^{.001}$ | $.75^{.009}$ | $.94^{.004}$ | $.39^{.006}$ | $.72^{.010}$ | $.37^{.012}$ | $.66^{.008}$ |
| **K**: F$^3$ with alignment | $\mathbf{.83^{.010}}$ | $\mathbf{.97^{.001}}$ | $\mathbf{.86^{.005}}$ | $\mathbf{.98^{.002}}$ | $\mathbf{.39^{.008}}$ | $.73^{.008}$ | $\mathbf{.45^{.015}}$ | $.72^{.003}$ |
| **L**: VFL w. graph/alignment | $.83^{.012}$ | $.97^{.001}$ | $.86^{.014}$ | $.98^{.002}$ | $.39^{.006}$ | $\mathbf{.74^{.009}}$ | $.45^{.015}$ | $\mathbf{.73^{.003}}$ |
| **M**: VFL w.o. pretrained local | $.77^{.02}$ | $.94^{.021}$ | $.77^{.014}$ | $.95^{.006}$ | $.34^{.014}$ | $.69^{.012}$ | $.35^{.008}$ | $.65^{.014}$ |

Table 3: Impact of learning graph across different alignment settings. $^\star$ Some references of rows are with respect to Table 2.

| | | METR-LA | | PEMS-BAY | | PMU-B | | PMU-C | |
|---|---|---|---|---|---|---|---|---|---|
| | | F1 | AUC | F1 | AUC | F1 | AUC | F1 | AUC |
| **No Align** | No Graph | $.768^{.009}$ | $.957^{.004}$ | $.738^{.012}$ | $.935^{.001}$ | $.381^{.008}$ | $.711^{.006}$ | $.342^{.008}$ | $.636^{.010}$ |
| | Given Graph | $.763^{.020}$ | $.957^{.007}$ | $.742^{.024}$ | $.942^{.005}$ | - | - | - | - |
| | $k$-NN Graph | $.715^{.015}$ | $.952^{.004}$ | $.695^{.013}$ | $.934^{.004}$ | $.372^{.001}$ | $.711^{.013}$ | $.404^{.016}$ | $.68^{.014}$ |
| | ICDF | $.798^{.009}$ | $.963^{.004}$ | $.755^{.009}$ | $.943^{.001}$ | $.387^{.003}$ | $.734^{.015}$ | $.403^{.020}$ | $.663^{.018}$ |
| **Align** | No Graph | $.813^{.009}$ | $.970^{.002}$ | $.846^{.008}$ | $.977^{.001}$ | $.386^{.009}$ | $.725^{.012}$ | $.386^{.008}$ | $.694^{.005}$ |
| | Given Graph | $.828^{.007}$ | $.974^{.001}$ | $.854^{.003}$ | $.977^{.001}$ | - | - | - | - |
| | $k$-NN Graph | $.803^{.020}$ | $.968^{.002}$ | $.855^{.003}$ | $.973^{.002}$ | $.378^{.002}$ | $.718^{.015}$ | $.418^{.007}$ | $.702^{.009}$ |
| | ICDF | $\mathbf{.835^{.010}}$ | $\mathbf{.975^{.001}}$ | $\mathbf{.860^{.005}}$ | $\mathbf{.980^{.002}}$ | $\mathbf{.390^{.008}}$ | $\mathbf{.734^{.008}}$ | $\mathbf{.451^{.015}}$ | $\mathbf{.725^{.003}}$ |

component. In particular, for each alignment setting, we demonstrate the impact on performance with (a) not using a graph; (b) using a graph given by the domain experts; (c) learning the graph structure using a $k$-**NN** baseline (Fatemi et al., 2021); and (d) learning the graph structure using **ICDF**. The reported numbers suggest that regardless of whether the model performs alignment, graph learning always improves performance.

**Remark.** The $k$-**NN** baseline ($k = 10$) is implemented following the description in (Fatemi et al., 2021). Specifically, during training, we generate a local graph for each batch for node features **X** via a symmetrization of $\tilde{\mathbf{A}} = k\text{-}\mathbf{NN}(\text{MLP}(\mathbf{X}))$ which (1) feeds the node features through a MLP neural block; and (2) draws an edge between each node and its $k$ nearest neighbors where the neighborhood is defined using the cosine similarity on the space of MLP-projected feature vectors.

# 7 CONCLUSIONS

In this paper, we study federated feature fusion, which presents a less addressed scenario of federated learning where data owners or clients need to customize their own local models to accommodate different sets of (federated) features. Unlike federated learning, the clients need to learn their own model separately in isolation and only communicate their local feature representations afterwards. We motivate the practicality of federated feature fusion scheme with a power grid example and propose a local–global model framework for it. Two important components of the framework are the alignment of the data representations produced by local models and the learning of the global model by using a graph neural network. Comprehensive experiments suggest the feasibility of federated feature fusion and the effectiveness of the framework.

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

## A  RELATED WORK

The concept of federated learning was first coined by McMahan et al. (2017) and it has attracted surging interests since. A form of distributed optimization, federated learning is faced with data challenges beyond conventional assumptions and puts communication efficiency and data privacy as primary concerns. Recent surveys (Yang et al., 2019a; Li et al., 2019; Kairouz et al., 2019) comprehensively study the subject, review systems and infrastructures, and suggest open problems.

The typical setting of federated learning is that data sets across owners share the same feature space but differ in samples. Besides this horizontal partitioning of the data matrix, a vertical partitioning was studied by Hardy et al. (2017); Nock et al. (2018); Heinze et al. (2014; 2016), wherein features are split across owners instead. This setting bears resemblance to our federated feature fusion scenario, but a crucial distinction is that existing methods for vertical federated learning all perform joint training. In the referenced work, to preserve privacy, encrypted data or randomly projected data are communicated among data owners as well as a central coordinator. Such an approach incurs demanding communication for many owners. Recently, Chen et al. (2020) study a different model, whose parameters are distributed among owners as well as a central server. The part of the model corresponding to an owner bears resemblance to our local models; but they are not local models since they are not independently trained by using local data. Another work along a similar direction is conducted by Hu et al. (2019), but the global model has no parameters; it is merely a sum of the local outputs followed by activation (e.g., sigmoid for classification).

Our framework learns parameter matrices to align local representations. Such alignments similarly appear in model fusion, where a number of models are fused together into a common model through aligning model parameters (Yurochkin et al., 2019a). In the context of deep learning, if the neural networks come from the same model family, their weights can be matched layerwise, even if the numbers of weights are different (Yurochkin et al., 2019b; Wang et al., 2020). The referenced work treats the problem as a bipartite graph matching, where the cost matrix is inferred from maximum a posteriori estimation. Then, the Hungarian algorithm (Kuhn, 1955) is applied to find the matching. In our work, instead we treat the permutation alignment as a differentiable parameterization with the help of Sinkhorn–Knopp (Sinkhorn & Knopp, 1967; Mena et al., 2018; Emami & Ranka, 2018), so that it can be learned end-to-end with other parameters of the global model.

Our framework also advocates learning a graph of data owners in the global model. Graph structure learning appears under various contexts. One field of study is probabilistic graphical models and casual inference, whereby a directed acyclic structure is learned. Gradient-based approaches in this context include Zheng et al. (2018); Yu et al. (2019); Lachapelle et al. (2020). On the other hand, a general graph may still be useful without resorting to causality. Recent approaches supporting GNN-based modeling include Kipf et al. (2018); Franceschi et al. (2019); Wu et al. (2020); Shang et al. (2021), wherein a graph structure is simultaneously learned together with the GNN parameters. The Gumbel trick (Jang et al., 2017; Maddison et al., 2017) is frequently used for differentiable parameterization, but in this paper we study a more economic alternative parameterization **ICDF**.

## B  PERMUTATION AMBIGUITY EXAMPLE FOR GRU

In Section 3, we discuss that one can arbitrarily permute the latent representations while keeping a local model fixed. Here, we give another example – the GRU. Let $x = \{x_1, x_2, \ldots, x_T\}$ be an input sequence. The embedding function $h = \text{embedding}(x)$ implemented as a GRU reads:

```
 1: function h = GRU({x_t}_{t=1,}^T)
 2:     h_0 = 0
 3:     for t = 1, ..., T do
 4:         z_t = sigmoid(W_z x_t + U_z h_{t-1} + b_z)
 5:         r_t = sigmoid(W_r x_t + U_r h_{t-1} + b_r)
 6:         n_t = tanh(W_n x_t + U_n(r_t ⊙ h_{t-1}) + b_n)
 7:         h_t = (1 - z_t) ⊙ h_{t-1} + z_t ⊙ n_t
 8:     end for
 9:     return h = h_T
10: end function
```

One can arbitrarily permute the elements of $h$ through manipulating the GRU parameters properly. To achieve $h[\mathrm{p}] = \mathrm{embedding}(x; \mathrm{p})$,

- the gate outputs and bias vectors ($z_t$, $r_t$, $n_t$, $h_t$, $b_z$, $b_r$, $b_n$) will need be permuted accordingly ($z_t[\mathrm{p}]$, $r_t[\mathrm{p}]$, $n_t[\mathrm{p}]$, $h_t[\mathrm{p}]$, $b_z[\mathrm{p}]$, $b_r[\mathrm{p}]$, $b_n[\mathrm{p}]$);

- the weight matrices attached to the input ($W_z$, $W_r$, $W_n$) will need to have their rows (i.e., output neurons) permuted ($W_z[\mathrm{p}, :]$, $W_r[\mathrm{p}, :]$, $W_n[\mathrm{p}, :]$); and

- the weight matrices attached to the hidden states ($U_z$, $U_r$, $U_n$) will need to have both their rows and columns permuted ($U_z[\mathrm{p}, \mathrm{p}]$, $U_r[\mathrm{p}, \mathrm{p}]$, $U_n[\mathrm{p}, \mathrm{p}]$).

## C  PROOFS AND ADDITIONAL RESULTS OF THEOREM 1, SECTION 5

### C.1  DISTRIBUTION OF GUMBEL SOFTMAX

The Gumbel softmax reparameterization trick (Jang et al., 2017; Maddison et al., 2017) works in the following manner. Let $\mathrm{Cat}(\boldsymbol{\pi})$ be the categorical distribution with probability vector $\boldsymbol{\pi}$ and let $\mathbf{g}$, of the same shape as $\boldsymbol{\pi}$, be a vector variable whose elements are i.i.d. $\sim \mathrm{Gumbel}(0, 1)$. Then,

$$y = \mathrm{softmax}\left(\frac{1}{\tau}\left(\log \boldsymbol{\pi} + \mathbf{g}\right)\right), \quad \tau > 0 \tag{11}$$

admits a distribution converging to $\mathrm{Cat}(\boldsymbol{\pi})$ when $\tau \to 0$. Hence, to sample $\mathrm{Ber}(\theta)$ approximately but differentiably, it suffices to let $\boldsymbol{\pi} = [\theta, 1 - \theta]^\top$ and use $y_1$ as the sample.

As preliminary, we consider the first entry $y_1$ of the random variable $y$ defined in (11) for the Gumbel softmax parameterization. Note that for any $\tau \neq 0$, $y_1$ is only approximately binary; the possible values of $y_1$ in fact span the entire interval $[0, 1]$. We derive the following CDF for $y_1$. Recall that for notational simplicity, $\theta$ denotes a scalar rather than a matrix.

**Theorem 2.** For all $\tau > 0$, $\theta \in (0, 1)$, and $t \in [0, 1]$, we have

$$\Pr(y_1 \leq t) = \frac{t^\tau(1 - \theta)}{t^\tau(1 - \theta) + (1 - t)^\tau \theta}. \tag{12}$$

**Proof.** We first consider the case $0 < t < 1$. Through simple algebraic manipulation, we obtain that $y_1 \leq t$ is equivalent to

$$g_1 - g_2 \leq \tau \log \frac{t}{1 - t} - \log \frac{\theta}{1 - \theta}. \tag{13}$$

Let $g_1 = -\log(-\log u)$ and $g_2 = -\log(-\log v)$, where $u$ and $v$ are independent and $\sim \mathcal{U}(0, 1)$. Then, (13) is equivalent to

$$v \geq u^M \quad \text{where} \quad M = \frac{t^\tau(1 - \theta)}{(1 - t)^\tau \theta}.$$

Therefore, by recalling that $u$ and $v$ are uniform in $[0, 1]^2$, we note that the probability that $v \leq u^M$ happens is the double integral

$$\Pr(v \geq u^M) = \int_0^1 \int_{u^M}^1 1 \, dv du.$$

This integral is nothing but

$$1 - \int_0^1 u^M \, du = \frac{M}{1 + M},$$

which completes the proof of (12). The cases of $t = 0$ or $1$ obviously hold by continuity.

## C.2 Proof of Theorem 1

We first consider the case when the distribution with cdf $F$ is finitely supported on $[a, b]$. Through simple algebraic manipulation, we obtain that $z \leq t$ is equivalent to $s \geq M$ where $M := F^{-1}(\theta) + \tau \log(t^{-1} - 1)$. If $t < \text{sigmoid}((F^{-1}(\theta) - b)/\tau)$, we see that $M > b$ and thus such $s$ can never occur. Similarly, if $t > \text{sigmoid}((F^{-1}(\theta) - a)/\tau)$, we see that $M < a$, which indicates that $s \geq M$ always happens. Otherwise, when $t$ is within the two extremes, the probability that $s \geq M$ happens is $1 - F(M)$, concluding the proof of (10).

The statement of the theorem regarding the case when the distribution is not finitely supported is obviously true.

To show that the distribution of $z$ converges to $\text{Ber}(\theta)$, let us first consider the scenario when the distribution with cdf $F$ is finitely supported. The cdf of $z$ (see (10)) is always continuous but it has three segments connected by two joints: $t_1 = \text{sigmoid}((F^{-1}(\theta) - b)/\tau)$ and $t_2 = \text{sigmoid}((F^{-1}(\theta) - a)/\tau)$. When $\tau \to 0$, the joint $t_1 \to 0$ and the joint $t_2 \to 1$ and thus the middle segment has a wider and wider support converging to $[0, 1]$. Hence, it suffices to consider only the middle segment. Further, with an analogous argument for other scenarios, it is also true that it suffices to consider only the third case of (10).

In this case, for any fixed $t < 1$ and when $\tau \to 0$, we have $\tau \log(t^{-1} - 1) \to 0$ and thus $\Pr(z \leq t) \to 1 - F(F^{-1}(\theta)) = 1 - \theta$. Meanwhile, we cannot push $\tau \to 1$ because then the limit of $\tau \log(t^{-1} - 1)$ is undefined. However, we know by definition that $\Pr(z \leq 1) = 1$. Hence, the continuous distribution of $z$ converges to a degenerate distribution $\Pr(z < 1) = 1 - \theta$ and $\Pr(z = 1) = 1$. This is the CDF of $\text{Ber}(\theta)$.

## D Tuning Guidance for Temperature $\tau$

Our tuning guidance for the temperature $\tau$ is motivated from a asymptotic convergence comparison between **ICDF** and Gumbel re-parameterization, which is featured in the theorem below.

**Theorem 3.** When $\tau$ is small,

$$\text{Bias}(y_1) = \frac{1}{6}\tau^2\pi^2\theta(1 - \theta)(1 - 2\theta) + O(\tau^4), \tag{14}$$

$$\text{Bias}(z) = \frac{1}{6}\tau^2\pi^2 F''(F^{-1}(\theta)) + O(\tau^4). \tag{15}$$

Moreover, when $F$ is the CDF of a normal variable $\sim \mathbb{N}(0, \sigma^2)$,

$$\text{Bias}(z) = -\frac{1}{6\sigma^2}\tau^2\pi^{\frac{3}{2}}\text{erf}^{-1}(2\theta - 1)e^{-(\text{erf}^{-1}(2\theta-1))^2} + O(\tau^4). \tag{16}$$

Its formal proof is detailed later in Appendix E.

Theorem 3 suggests that the **ICDF** method converges equally fast as does the Gumbel trick – both on the order of $O(\tau^2)$. On the other hand, the biases depend on $\theta$. Thus, one cannot set temperatures $\tau$ independently of the desired probability $\theta$ to equate the two biases. In practice, $\tau$ is a tunable hyper-parameter and a guidance on the tuning range is therefore necessary.

To begin, we use a subscript to distinguish the two temperatures – $\tau_\text{g}$ for the Gumbel trick and $\tau_\text{i}$ for the **ICDF** method – and write, based on (14) and (16) and ignoring the high order terms,

$$\frac{\text{Bias}(y_1)}{\text{Bias}(z)} \simeq \frac{\tau_\text{g}^2\sigma^2}{\tau_\text{i}^2}r(\theta) \quad \text{where} \quad r(\theta) = \frac{\sqrt{\pi}\theta(1 - \theta)(2\theta - 1)}{\text{erf}^{-1}(2\theta - 1)e^{-(\text{erf}^{-1}(2\theta-1))^2}}.$$

Note that $r(\theta)$ is symmetric around $\theta = \frac{1}{2}$, is concave, attains maximum $\frac{1}{2}$ when $\theta = \frac{1}{2}$, and attains minimum 0 when $\theta = 0, 1$. Hence, if $\tau_\text{g} = \tau_\text{i}$ and $\sigma = \sqrt{2}$, the bias of the Gumbel trick is (approximately) smaller than that of the icdf method. On the other hand, for a $\sigma > \sqrt{2}$, there exist $\widetilde{\theta}_1 < \widetilde{\theta}_2$ such that $\sigma^{-2} = r(\widetilde{\theta}_1) = r(\widetilde{\theta}_2)$ and that $\text{Bias}(y_1) \gtrsim \text{Bias}(z)$, whenever $\theta \in [\widetilde{\theta}_1, \widetilde{\theta}_2]$. For example, when $\sigma \approx 2.5$, on the interval $\theta \in [0.01, 0.99]$, the bias of the Gumbel trick is (approximately) greater than that of the icdf method.

Based on the foregoing, a practical guide is to use the same tuning range of $\tau$ for the **ICDF** method as for the Gumbel trick. A small change of $\sigma$ (e.g., $\sqrt{2}$ versus 2.5) will entirely flip the landscape of the bias comparison between the two methods. Because the tuning range is much wider than the change of $\sigma$, for simplicity it suffices to fix $\sigma = 1$.

## E    PROOF OF THEOREM 3 AND ADDITIONAL RESULTS

By the definition of bias, we have

$$\text{Bias}(x) = \mathbb{E}[x] - \theta \quad \text{where} \quad \mathbb{E}[x] = \int_0^1 t \, d \Pr(x \le t) = 1 - \int_0^1 \Pr(x \le t) \, dt.$$

Therefore, for Gumbel softmax,

$$\text{Bias}(y_1) = 1 - \theta - \int_0^1 \frac{t^\tau (1 - \theta)}{t^\tau (1 - \theta) + (1 - t)^\tau \theta} \, dt,$$

and for icdf with any $F$,

$$\text{Bias}(z) = \int_0^1 F(F^{-1}(\theta) + \tau \log(t^{-1} - 1)) \, dt - \theta.$$

We now prove Theorem 3 in a few parts.

**Proof of (15).** Let $s = F^{-1}(\theta)$ and perform a change of variable $m = \log(t^{-1} - 1)$. Then,

$$\text{Bias}(z) = \int_0^1 [F(s + \tau m) - F(s)] \, dt = \int_{-\infty}^\infty [F(s + \tau m) - F(s)] \frac{e^m}{(1 + e^m)^2} \, dm.$$

We perform Taylor expansion of $F$ around $s$ and obtain

$$F(s + \tau m) - F(s) = \sum_{n=1}^\infty \frac{F^{(n)}(s)}{n!} \tau^n m^n.$$

Therefore,

$$\text{Bias}(z) = \sum_{n=1}^\infty \frac{F^{(n)}(s)}{n!} \tau^n \int_{-\infty}^\infty \frac{m^n e^m}{(1 + e^m)^2} \, dm$$

Each integral term is finite and the odd terms vanish because the integrands are odd functions. Thus, for small $\tau$, we are left with

$$\text{Bias}(z) = \frac{F''(s)}{2} \tau^2 \int_{-\infty}^\infty \frac{m^2 e^m}{(1 + e^m)^2} \, dm + O(\tau^4).$$

The definite integral evaluates to $\frac{\pi^2}{3}$; we therefore conclude the proof.

**Proof of (16).** Equation (16) is straightforward by substuting

$$F''(s) = -\frac{s}{\sigma^3 \sqrt{2\pi}} e^{-\frac{s^2}{2\sigma^2}} = -\frac{\text{erf}^{-1}(2\theta - 1)}{\sigma^2 \sqrt{\pi}} e^{-(\text{erf}^{-1}(2\theta - 1))^2}.$$

into (15).

**Proof of (14).** To simplify notation, let $\beta = \theta/(1 - \theta)$ and perform a change of variable $m = \log(t^{-1} - 1)$. Then,

$$\int_0^1 \frac{t^\tau (1 - \theta)}{t^\tau (1 - \theta) + (1 - t)^\tau \theta} \, dt = \int_0^1 \frac{dt}{1 + \beta e^{m\tau}} = \int_{-\infty}^\infty \frac{1}{1 + \beta e^{m\tau}} \frac{e^m}{(1 + e^m)^2} \, dm.$$

Denote $h(\tau, m) = [1 + \beta e^{m\tau}]^{-1}$. Treating $h$ a function of $\tau$ and performing Taylor expansion around zero, we obtain

$$h(\tau, m) = \sum_{n=0}^\infty \frac{h^{(n)}(0, m)}{n!} \tau^n.$$

Therefore,

$$\int_0^1 \frac{t^\tau(1-\theta)}{t^\tau(1-\theta)+(1-t)^\tau\theta}\,dt = \sum_{n=0}^\infty \frac{\tau^n}{n!}\int_{-\infty}^\infty h^{(n)}(0,m)\frac{e^m}{(1+e^m)^2}\,dm.$$

In a moment, we will show that for all $n$,

$$h^{(n)}(0,m) = C_n m^n \quad \text{where } C_n \text{ is independent of } m. \tag{17}$$

Suppose that (17) holds. Then, each integral term is finite and the odd terms vanish, because the integrands are odd functions. Therefore, for small $\tau$, we are left with

$$\int_0^1 \frac{t^\tau(1-\theta)}{t^\tau(1-\theta)+(1-t)^\tau\theta}\,dt = C_0 \int_{-\infty}^\infty \frac{e^m}{(1+e^m)^2}\,dm + C_2\frac{\tau^2}{2}\int_{-\infty}^\infty \frac{m^2 e^m}{(1+e^m)^2}\,dm + O(\tau^4).$$

By calculating

$$C_0 = h(0,m) = [1+\beta]^{-1} = 1-\theta, \qquad C_2 = h''(0,m) = -\theta(1-\theta)(1-2\theta),$$
$$\int_{-\infty}^\infty \frac{e^m}{(1+e^m)^2}\,dm = 1, \qquad \int_{-\infty}^\infty \frac{m^2 e^m}{(1+e^m)^2}\,dm = \frac{\pi^2}{3},$$

we conclude that

$$\text{Bias}(y_1) = \frac{\tau^2\pi^2\theta(1-\theta)(1-2\theta)}{6} + O(\tau^4).$$

It remains to prove (17). We suppress the argument on $m$ and write $g(\tau) = 1+\beta e^{m\tau}$ and $h(\tau) = g(\tau)^{-1}$. By Faà di Bruno's formula,

$$h^{(n)}(0) = \left.\left(\frac{1}{g(\tau)}\right)^{(n)}\right|_{\tau=0} = \sum_{k=1}^n \frac{(-1)^k k!}{g(0)^{k+1}}\cdot B_{n,k}\Big(g'(0), g''(0), \ldots, g^{(n-k+1)}(0)\Big),$$

where $B_{n,k}$ is the Bell polynomial. Clearly, $g(0) = 1+\beta$ and $g^{(r)}(0) = \beta m^r$ for all $r > 0$. Hence, $B_{n,k}$ is a multiple of $m^n$. Therefore, $h^{(n)}(0)$ is a multiple of $m^n$.

### E.1 ADDITIONAL RESULT REGARDING THE BIAS

Theorem 3 states results for a small temperature $\tau$. The purpose is to understand the limiting behavior of the bias. Here, we give an additional result for any $\tau > 0$. It states that the biases of the two sampling approaches have the same sign. This result is a nontrivial extension of Theorem 3 and requires a different proof technique.

**Theorem 4.** For any $\tau > 0$,

$$\text{Bias}(y_1) > 0 \text{ when } \theta < \tfrac{1}{2}, \quad \text{Bias}(y_1) = 0 \text{ when } \theta = \tfrac{1}{2}, \quad \text{Bias}(y_1) < 0 \text{ when } \theta > \tfrac{1}{2}. \tag{18}$$

Moreover, if $F'(x)$ (that is, the pdf) is even and is increasing when $x < 0$, then

$$\text{Bias}(z) > 0 \text{ when } \theta < \tfrac{1}{2}, \quad \text{Bias}(z) = 0 \text{ when } \theta = \tfrac{1}{2}, \quad \text{Bias}(z) < 0 \text{ when } \theta > \tfrac{1}{2}. \tag{19}$$

We prove Theorem 4 in two parts.

**Proof of (18).** Consider

$$\text{Bias}(y_1) = \int_0^1 g(t,\theta)\,dt \quad \text{where} \quad g(t,\theta) = 1-\theta - \frac{t^\tau(1-\theta)}{t^\tau(1-\theta)+(1-t)^\tau\theta}.$$

With a brute-force calculation, we have

$$g(t,\theta) + g(1-t,\theta) = \frac{[(1-t)^\tau - t^\tau]^2\theta(1-\theta)(1-2\theta)}{[t^\tau(1-\theta)+(1-t)^\tau\theta][(1-t)^\tau(1-\theta)+t^\tau\theta]}.$$

All terms on the right-hand side are positive, except $1-2\theta$. Therefore, when $\theta < \tfrac{1}{2}$, $g(t,\theta)+g(1-t,\theta) > 0$ and hence

$$\text{Bias}(y_1) = \int_0^1 \frac{g(t,\theta)+g(1-t,\theta)}{2}\,dt > 0.$$

The other cases ($\theta > \frac{1}{2}$ and $\theta = \frac{1}{2}$) are similarly proved.

**Proof of** (19)**.** Consider

$$\text{Bias}(z) = \int_0^1 h(t, \theta)\, dt - \theta \quad \text{where} \quad h(t, \theta) = F(F^{-1}(\theta) + \tau \log(t^{-1} - 1)).$$

We have

$$h(1 - t, \theta) = F(F^{-1}(\theta) - \tau \log(t^{-1} - 1)).$$

To simplify notation, let $F^{-1}(\theta) = s$ and $\tau \log(t^{-1} - 1) = a$. Then, $h(t, \theta) = F(s + a)$ and $h(1 - t, \theta) = F(s - a)$. Let us first consider the case $s < 0$ and $a > 0$. We see that

$$F(s + a) - F(s) = \int_s^{s+a} F'(m)\, dm \quad \text{and} \quad F(s) - F(s - a) = \int_{s-a}^s F'(m)\, dm.$$

For any $b > 0$, if $s + b < 0$, then by monotonicity, $F'(s + b) > F'(s - b)$. On the other hand, if $s + b \geq 0$, then $F'(s + b) = F'(-s - b) > F'(s - b)$. In both cases, the right integral is always smaller than the left integral. In other words,

$$F(s + a) + F(s - a) > 2F(s).$$

In fact, the above inequality is also established when $s < 0$ and $a < 0$. Therefore, whenever $s < 0$,

$$\int_0^1 h(t, \theta)\, dt = \int_0^1 \frac{h(t, \theta) + h(1 - t, \theta)}{2}\, dt > \int_0^1 F(F^{-1}(\theta))\, dt = \theta.$$

That is, $\text{Bias}(z) > 0$. Other cases ($s = F^{-1}(\theta) > 0$ and $s = F^{-1}(\theta) = 0$) are similarly proved.

### E.2 Empirical Comparison between Gumbel and **ICDF** re-parameterization

Extending the last experiment in Section 6, Table 4 summarizes the time and memory consumption during the training of global models on the four data sets. The results indicate that our developed **ICDF** re-parameterization is more economic than the Gumbel-Softmax approach.

Table 4: Time and memory consumption of $F^3$ (five epochs) with respect to **ICDF** and Gumbel-Softmax re-parameterization. Time is in seconds and memory is in MB.

| | METR-LA | | PEMS-BAY | | PMU-B | | PMU-C | |
|---|---|---|---|---|---|---|---|---|
| | Time | Memory | Time | Memory | Time | Memory | Time | Memory |
| **Gumbel-Softmax** | 87.89 | 832.38 | 270.52 | 1896.11 | 42.40 | 348.39 | 84.89 | 1119.13 |
| **ICDF** | 79.69 | 568.24 | 157.93 | 1167.19 | 30.16 | 322.59 | 54.07 | 894.63 |

## F Data Set Description and Preprocessing

**METR-LA** and **PEMS-BAY.** These are traffic data sets (MIT licensed) used by Li et al. (2018). The former was collected from loop detectors in the highway of Los Angles, CA (Jagadish et al., 2014) and the latter was collected by the California Transportation Agencies Performance Measure System. Both data sets recorded several months of data at the resolution of five minutes. The network graphs are available, which were constructed by imposing a radial basis function on the pairwise distance of sensors at a certain cutoff.

The data sets were originally prepared for forecasting tasks and hence no labeling information exists. We adapt the data for classification. Specifically, we split the time series on the hour, forming hourly windows. We label each window as whether or not it corresponds to rush hour. For proof of concept, we specify 07:00–10:00 and 16:00–19:00 as rush hour and the others non-rush hour. We note that in the original data sets, one of the attributes is time. We remove this attribute to avoid triviality and retain only the speed attribute.

The specification of rush hours may not be highly accurate, but it is a sensible practice to cope with the nonexistence of labeling information. Intuitively, the signal of rush hour comes from reduced

traffic speed, but not every location of the network experiences traffic jam. Hence, the diverse traffic patterns inside the same time window under a single label causes nontrivial challenges for local models to discern. Therefore, the need of a global consensus model is justified and it fits well the federated inference scenario.

**PMU-B** and **PMU-C.** These are proprietary data sets coordinately provided by multiple data owners of the U.S. power grid. No personally identifiable information is present. The suffixes B and C indicate the interconnects of the grid. The data sets come with thousands of annotated grid events spanning a period of two years; they form the classification labels. Many variables (attributes) of the grid condition are recorded; we select only the voltage magnitude and the current magnitude, because they appear to be the strongest signals for event detection based on domain knowledge, and also because more data are available for these two variables. The grid topology is not available.

For each event, we select a one-second window from the three-minute window that covers the approximate annotated event time, based on the largest z-score. We retain a sampling frequency of 30Hz, even though some data are 60Hz. Furthermore, a large amount of data are missing in the raw data. We impute the series by using `pandas.DataFrame.interpolate(method = 'linear', limit_direction = 'both')` from the Python `pandas` package. This way, a windowed series is complete if it ever has raw data. Even so, many series are entirely empty, which corresponds to the scenario illustrated by Figure 1. Classes in these two data sets are rather skewed. For PMU-B, we remove a class that consists of only one data point and for PMU-C, we combine classes that contain fewer than 24 data points into a single class.

## G  EXPERIMENT DETAILS

The experiments are conducted on one x86 node of a computing cluster with one a100 NVIDIA GPU. The compute node has eight Intel cores and 128GB memory. For each data set, we perform a 70/10/20 random split for training, validation, and testing, respectively.

For local models, we use LSTM with the same hyperparameters: one hidden layer whose hidden dimension is 16 and the maximum number of epochs $= 200$. We pre-train the local models and freeze their parameters afterward. We train each global model for a maximum of 500 epochs and use early stopping according to the validation loss, with a patience of 50 epochs. For the GNN global model, we use a 2-layer GCN with skip connections. The hidden dimension is set at 8 and we select the learning rate from $\{0.01, 0.001\}$. For missing data, we impute the node features by using zero.

## H  SOFT AND HARD FEATURE ALIGNMENT

Feature alignment can be achieved in two manners. The first approach is a soft alignment, which treats each $P^i$ a free parameter matrix to optimize. Such an alignment softens the one-to-one correspondence in the permutation constraint; i.e., each feature in the source can have a weighted correspondence to each of the features in the target. That is the way we used in the main paper.

An alternative approach is a hard alignment, which treats each $P^i$ a permutation matrix. Learning permutation matrices is challenging, however, because they correspond to combinatorial structures and are unsuitable for gradient-based training. We follow Mena et al. (2018); Emami & Ranka (2018) and relax $P^i$ by a doubly stochastic matrix, which can be differentiably parameterized by the Sinkhorn–Knopp algorithm Sinkhorn & Knopp (1967). Specifically, starting from a nonnegative square matrix $K_0$ and column vectors $r_0 = c_0 = \mathbf{1}$ of matching lengths, define the sequence

$$c_{j+1} = \mathbf{1} \oslash (K_0^T r_j) \text{ and } r_{j+1} = \mathbf{1} \oslash (K_0 c_j), \quad \text{for } j = 0, 1, \dots \tag{20}$$

Then, under a mild condition, $K_j := \operatorname{diag}(r_j) K_0 \operatorname{diag}(c_j)$ converges to a doubly stochastic matrix. We truncate the sequence at the $T$th step and treat $K_T$ as an approximation of $P^i$.

Despite the advocation by Mena et al. (2018); Emami & Ranka (2018), we obtain the following convergence result of Sinkhorn–Knopp, which reveals no free lunch.

Table 5: Different approaches to feature alignment.

| | METR-LA | | PEMS-BAY | | PMU-B | | PMU-C | |
|---|---|---|---|---|---|---|---|---|
| | F1 | AUC | F1 | AUC | F1 | AUC | F1 | AUC |
| soft alignment | .835 | .975 | .860 | .980 | .390 | .734 | .451 | .725 |
| hard alignment | .839 | .973 | .855 | .976 | .390 | .737 | .429 | .721 |

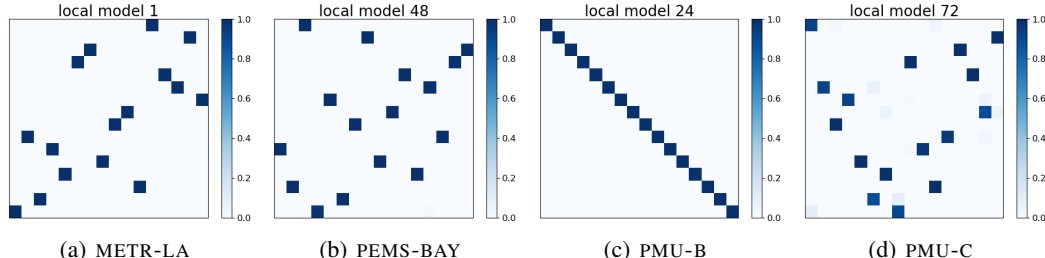

(a) METR-LA  (b) PEMS-BAY  (c) PMU-B  (d) PMU-C

Figure 4: Examples of learned permutation matrices ($K_T$). The plots clearly show patterns of a permutation matrix: there is one and only one significant value per row and per column. Because of the slow convergence, we attribute the desirable results of $K_T$ (at a small $T$) to the success of the learning of $K_0$. Note also interestingly that a learned permutation may be the identity mapping.

**Theorem 5** (informal). Under a condition of $K_0$, there exists a positive integer $J$ and a constant $C_J$ such that for all $j \geq J$,

$$\left\| \begin{bmatrix} K_j^T \mathbf{1} \\ K_j \mathbf{1} \end{bmatrix} - \begin{bmatrix} \mathbf{1} \\ \mathbf{1} \end{bmatrix} \right\| \leq C_J (1 + \sigma_2^2) \sigma_2^{2(j-J)},$$

where $\sigma_2 \leq 1$ is the second largest singular value of the limit of $K_j$.

Since this is not the focus of this paper, we omit the rigorous analysis of this theorem. The result suggests that for a desirable limit being a permutation matrix, whose $\sigma_2 = 1$, the error $O(\sigma_2^{2j})$ does not drop. In practice, to expect for an approximate permutation matrix, $\sigma_2 \approx 1$ and the convergence is exceedingly slow. The practical usefulness of (20) depends on the learned quality of $K_0$.

The soft and hard alignment approaches have pros and cons. The hard approach maintains the correspondence of each feature dimension of the latent vectors while the soft approach . Maintaining the dimension correspondence is an advantage, especially for local models that produce disentangled latent representations Higgins et al. (2018), because each feature dimension is equipped with a semantic meaning that controls a certain aspect of the data. On the other hand, the soft approach is more straightforward and the hard approach is based on an algorithm that barely converges. In practice, we observe that two approaches decisively similar performance. Due to space limitation we took the simpler approach and presented only the soft version in the main paper, but here we list the results for both approaches in Table 5, and we also visualize the hard alignment matrix learned on each dataset to help readers understand the feature alignment (Figure 4).

