# OpenReview forum: "Merging Models Pre-Trained on Different Features with Consensus Graph"
_ICLR.cc/2023/Conference — Submitted to ICLR 2023_

### Official Review · Reviewer_Szi9 · 2022-10-16

**Confidence:** 3
**Correctness:** 2
**Technical Novelty And Significance:** 3
**Empirical Novelty And Significance:** 3
**Recommendation:** 5

**Clarity, Quality, Novelty And Reproducibility:**

Clarity: Good. This paper is clearly organized and easy to follow. I appreciate the authors' efforts to justify the studies setting using practical examples.

Quality: The technical quality has some minor flaws and ambiguity, such as label sharing, feature alignment. See Weaknesses.

Novelty: Although the studied training setting is new, the model design highly resembles CNFGNN. I would like to see some discussions between $F^3$ and CNFGNN to justify its novelty.

Reproducibility: The paper is not submitted with codes. However, I think that the paper clearly describes the methodology such that researchers with adequate background can reproduce it.

**Strength And Weaknesses:**

Strengths:
- The studied problem is valid and novel. VFL does have the problem of high synchronization cost, while in practice, parties may indeed own heterogeneous features. Moreover, the authors justify the motivation using real-world examples clearly. I appreciate that.
- The paper is clearly written and well-organized. I am able to understand the motivation, key challenges, design components, etc. of this work.

Weaknesses:
- Missing related work. Although this paper tackles the setting of independent training and joint inference, the formulation of federated training over time-series data with an underlying graph is similar to CNFGNN (Meng et al. 2021). The authors should discuss the relationship of this work and CNFGNN, especially over the model design. From my perspective, both CNFGNN and $F^3$ use separate temporal feature extractor, and use GNN to aggregate the temporal features.
- Not clear why the $\mathbf{P}$ does feature alignment. From my perspective, the matrices $\mathbf{P}$ are not regularized (e.g. requiring local features to carry similar distributions) , and without specific parameterization (e.g. permutation matrices). Therefore, whether and why $\mathbf{P}$ actually does feature alignment are at doubt. The effectiveness of introducing $\mathbf{P}$ can easily come from introducing additional trainable parameters, instead of feature alignment. Some justifications are needed.
- How to train the global GCN and graph structure learning is not clear. I assume that different parties own different labels, as indicated by Section 2, "some local models may report event whereas others report normal". Therefore, it seems that to train the global GNN, local labels have to be somehow passed to the global server. However, this seems to violate the privacy requirement of FL. Some justifications are needed on this point.
- Efficiency advantage over asynchronous VFL is not clear. The motivation of $F^3$ instead of VFL is that VFL requires synchronization, which is not efficient. However, there are works on asynchronous VFL (e.g. FDML (Hu et al. 2019)) that breaks the synchronization requirement at the cost of some feature staleness. The efficiency advantage of $F^3$ over asynchronous VFL is thus not clear and requires justification.

C. Meng et al. Cross-Node Federated Graph Neural Network for Spatio-Temporal Data Modeling. KDD 2021

Y. Hu et al. FDML: A Collaborative Machine Learning Framework for Distributed Features. KDD 2019


**Summary Of The Paper:**

This paper studies a problem in federated learning where local models are independently trained, but are aggregated to perform inference. This setting is a more practical setting than vertical federated learning (VFL) as it does not require expensive collaboration. To solve two challenges in this setting, feature misalignment and finding interdependency, the authors propose $F^3$. $F^3$ uses trainable modules to align the features generated from local models. $F^3$ then leverages graph neural network (GNN) and graph structure learning to capture interdependencies between local models. Experimental results on spatio-temporal data show the effectiveness of $F^3$.

**Summary Of The Review:**

My main concerns on this paper lies in 1) Lacking discussion to a related work CNFGNN with similar model design, and 2) Some ambiguity regarding technical correctness (ref. weaknesses). I recommend a weak reject at this stage. I am open to raising the recommendation if the authors clarify my doubts.

---

### Official Review · Reviewer_iTHt · 2022-10-25

**Confidence:** 4
**Correctness:** 3
**Technical Novelty And Significance:** 2
**Empirical Novelty And Significance:** 2
**Recommendation:** 5

**Clarity, Quality, Novelty And Reproducibility:**

The paper is clearly written and easy to follow. The proposed method still focus on solving the problem in vertical federated learning. The novelty is limited and the progress is not clearly validated since many necessary analysis and comparison are missing.

**Strength And Weaknesses:**

Strength:
1. The studies problems are important and practical.
2. The challenges of the studied setting are well summarized. The idea of consensus graph is practical when iterative training synchronicity is not possible.

Weakness:
1. The studies problem still lie in the vertical federated learning and the related work on the vertical federated learning does not cover the enough works, e.g, [1-4].
2. The convergence analysis and privacy preserving analysis of federated learning algorithms are important and both of them are missing in the paper
3. In the experiments, the paper need to be clearly compared with existing efforts in vertical federated learning.  The baseline G is a federated learning method which shows strong performance. First, the details of this method need to be included in the paper. Second, the good performance of the simple federated leanring baseline raise one question: if  the challenges regarding existing vertical federated learning in the introduction are over-claimed.
4. The analysis regarding  communication and training efficiency is missing. As demonstrated in the introduction, "Expensive coordination and synchronization among clients and the central server would be required, still imposing a strong constraint on synchronicity". To support the claim, the improvement in the synchronicity need to be provided and the comparison against  Vertical Federated Learning is also needed.



[1] Kewei Cheng, Tao Fan, Yilun Jin, Yang Liu, Tianjian Chen, Dimitrios Papadopoulos, and Qiang Yang. 2021. SecureBoost: A Lossless Federated Learning Framework. IEEE Intell. Syst.
[2] Fangcheng Fu, Yingxia Shao, Lele Yu, Jiawei Jiang, Huanran Xue, Yangyu Tao, and Bin Cui. 2021. VF2Boost: Very Fast Vertical Federated Gradient Boosting for Cross-Enterprise Learning. In Proceedings of the 2021 International Conference on Management of Data, SIGMOD 2021. ACM, 563–576
[3] Oscar Li, Jiankai Sun, Xin Yang, Weihao Gao, Hongyi Zhang, Junyuan Xie, Virginia Smith, and Chong Wang. 2022. Label Leakage and Protection in Two-party Split Learning. In International Conference on Learning Representations, ICLR 2022
[4] Shengwen Yang, Bing Ren, Xuhui Zhou, and Liping Liu. 2019. Parallel Distributed Logistic Regression for Vertical Federated Learning without Third-Party Coordinator. CoRR abs/1911.09824 (2019). arXiv:1911.09824

**Summary Of The Paper:**

The authors study how to learn an effective global model on private and decentralized datasets. Toward this end, the authors propose to use consensus graph to fuse feature representations for a final prediction. The experimental results on four real-life datasets are used to validate the effectiveness of the proposed framework.

**Summary Of The Review:**

The paper has its own merits but many necessary analysis and comparisons are missing. The comparisons against existing vertical federated learning need to be carefully conducted. The efficiency improvements need to be supported by empirical studies.  The convergence and privacy preserving analysis are needed. More details are listed in the Strength And Weaknesses section.

---

### Official Review · Reviewer_6rxh · 2022-10-25

**Confidence:** 3
**Clarity, Quality, Novelty And Reproducibility:** The concepts in the work are clear an…
**Correctness:** 4
**Technical Novelty And Significance:** 3
**Empirical Novelty And Significance:** 4
**Recommendation:** 8

**Strength And Weaknesses:**

The combination of feature alignment and graph consensus is a novel technique. The work has excellent ablation studies that concretely show the effectiveness of each aspect of the algorithm. One thing I want to note is that I think the datasets are uncommon in the FL community and it would be helpful to add brief descriptions of what the features and class labels are in the main text instead of the appendix.

**Summary Of The Paper:**

The paper proposes 'federated feature fusion' for application in the vertical federated learning paradigm using a combination of feature alignment and consensus graph. The technique aligns feature representations from pre-trained client models using a learnable alignment function and aggregates using a graph convolution network that learns the relational interactions of the different features to aggregate the global model. The algorithm is tested on four real-world time series datasets using LSTM models. Experimental results show good performance and are accompanied ablation studies.

**Summary Of The Review:**

The work proposes to use a graph convolution network to learn relational interaction between clients and uses an alignment technique to align the feature representations. The authors show good experimental results and ablation studies. Overall the work is a good contribution to federated learning.

---

### Official Review · Reviewer_2ATz · 2022-10-27

**Confidence:** 5
**Correctness:** 2
**Technical Novelty And Significance:** 2
**Empirical Novelty And Significance:** 2
**Recommendation:** 5

**Clarity, Quality, Novelty And Reproducibility:**

Clarity: Most parts of the paper are written in a clear way, except for the relation between VFL and the two constraints.

Quality: The quality of the paper can be further improved. I prefer to see a comparison between F^3 and existing FL/VFL studies that deal with model heterogeneity or use feature fusion techniques.

Novelty: The novelty is fair. But the overall contribution to the FL community is limited.

Originality: Good.


**Strength And Weaknesses:**

Strength:

1. The methodology part is written in a clear way and well addressed the main claims in the introduction part.

2. It is an interesting idea to study the VFL problem from the feature fusion perspective.



Weaknesses:

1. The relation between the two mentioned constraints and VFL is not explicit. As illustrated in the Introduction, the main constraints considered in this paper are the homogeneity and synchronicity problem and the main setting is VFL. However, I do not see any discussion on the constraints and VFL. There should be more discussion on how the constraints hinder the performance of existing VFL methods.

2. It is better to involve more components, e.g., loss functions, model parameters, and communication progress, in Figure 2 to make it clearer.

3. It concerns me whether it is secure to directly transmit data representations from the data owner to a central server, because there are some techniques that can reconstruct raw samples from their representations.

4. It is not persuasive enough to use representation alignment and graph convolution. Section 4,5 do not relate them to VFL or existing VFL methods, and it seems that the two techniques are designed for a certain type of dataset.

5. There lacks a comparison to existing FL methods, especially the following works that utilize the graph to model the relationships among clients, e.g. graph-based aggregation [1], address the model heterogeneity problems, e.g. distillation-based fusion [2], gradient-wise aggregation [3], prototype-wise aggregation [4], and/or fuse representations output by various pre-trained models, e.g. pre-trained model-based aggregation[5].

[1] Personalized Federated Learning With A Graph. In IJCAI 2022.

[2] Ensemble Distillation for Robust Model Fusion in Federated Learning. In NeurIPS 2020.

[3] HeteroFL: Computation and communication efficient federated learning for heterogeneous clients. In ICLR 2020.

[4] FedProto: Federated prototype learning across heterogeneous clients. In AAAI 2022.

[5] Federated Learning from Pre-Trained Models: A Contrastive Learning Approach. arXiv 2022.


**Summary Of The Paper:**

This paper considers vertical federated learning settings. It addresses the homogeneity and synchronicity problems by proposing a federated feature fusion (F^3) framework and learning a consensus graph among local owners. Experiments on four real-world time series datasets demonstrate the effectiveness of F^3.

**Summary Of The Review:**

Please refer to the comments in Strengths and Weaknesses.

---

### Decision · Program_Chairs · 2023-01-20

**Decision:**

Reject

**Justification For Why Not Higher Score:**

Limited contribution to the FL community.

**Justification For Why Not Lower Score:**

N/A

**Metareview: Summary, Strengths And Weaknesses:**

This paper presents an approach to federated feature fusion, in which clients learn local models in isolation on data of different modalities corresponding to the same examples, then in one round of communication send their local features to a server for fusion.  Specifically it proposes to learn alignment matrices for each client to globally align the representations, then use a graph neural network to fuse the representations before making the final prediction on the server.  The reviewers found that although the method is novel, the contribution to the FL community is limited,  and it is unclear under what conditions the method converge meaningfully (i.e., will be able to fuse the local models successfully).